# ESKD from primary glomerulonephritis: Incidence and post KRT survival: A national cohort study in the UK

Stephen A Roberts[1‡], Omar Ragy[2,3‡*], Seb Bate[2], Anna Casula[4], Sandip Mitra[2,3], Jonathan Barratt[5], Anirudh Rao[6], Durga A.K. Kanigicherla[2,3], On behalf of the GN-UNITY Study group[¶]

1 Centre for Biostatistics, School of Health Services, University of Manchester, Manchester Academic Health Science Centre, Manchester, United Kingdom, 2 Manchester University Hospitals NHS Foundation Trust, Manchester Academy of Health Sciences Centre, Manchester, United Kingdom, 3 Division of Cardiovascular Sciences, University of Manchester, Manchester, United Kingdom, 4 UK Kidney Association, UK Renal Registry, Bristol, United Kingdom, 5 Department of Cardiovascular Sciences, University of Leicester, Leicester, United Kingdom, 6 Department of Nephrology and Transplantation, Liverpool University Hospitals NHS Foundation Trust, Liverpool, United Kingdom

‡ Joint first authors on this work.
¶ Membership of the GN-UNITY Study group is listed in the Acknowledgments.
* omar.ragy@mft.nhs.uk

## Abstract

### Background

Primary glomerulonephritis (pGN) is a significant contributor to end-stage kidney disease (ESKD) and presents a significant healthcare burden. However, the epidemiology of ESKD-pGN in the UK remains poorly characterised.

### Methods

We analysed UK Renal Registry data for adults initiating kidney replacement therapy (KRT) between 1999 and 2019, with follow-up to December 2021. The incident cohort included patients starting KRT from 2008 onwards, when national population coverage was achieved, reflecting incidence of KRT initiation due to primary glomerulonephritis rather than overall disease incidence. Incidence rates were normalised to population and analysed using multivariable Poisson regression. Overall patient and graft survival were assessed using Cox regression models with time-dependent covariates, adjusted for demographic variables. Patients managed conservatively were not captured, limiting conclusions to advanced disease requiring KRT. Adult polycystic kidney disease (ADPKD) served as a reference disease.

### Results

The cohort included 10,308 pGN and 8,302 ADPKD patients. Median age at KRT start in pGN was 54 years, with 71.6% male. Overall incidence of KRT from pGN was

**Data availability statement:** The data used in this study were obtained from the UK Renal Registry (UKRR) under a standard data access request, in accordance with UKRR governance procedures. The authors do not have legal authority to publicly distribute the individual-level registry data due to patient confidentiality and data protection regulations. The data underlying this study were obtained from the UK Renal Registry (UKRR), a national registry of patients receiving kidney replacement therapy in the United Kingdom. The dataset comprises de-identified individual-level data, including demographic characteristics, primary renal diagnosis (coded using ERA–EDTA classifications), treatment modality, transplantation events, and outcome data (including death and graft failure). We, as authors, accessed these data following approval of a standard data access application through the UK Renal Registry governance framework. Permission was granted for use of the dataset for the purposes of this specific research study. Due to legal, ethical, and data protection restrictions related to the UKRR policy, we are not permitted to publicly share the underlying individual-level data. We, as authors, confirm that we did not receive any special privileges in accessing the data that other qualified researchers would not have. Access is available to bona fide researchers through the UK Renal Registry's formal data request process. Details regarding application procedures, governance requirements, and data access policies are available on the UKRR website. Researchers wishing to access the data may apply directly to the UK Renal Registry via: UK Renal Registry, UK Kidney Association. All details pertaining to accessing UKRR data is on this page https://www.ukkidney.org/audit-research/how-access-data/ukrr-data.

**Funding:** This study was supported through a research grant from CSL Vifor.

**Competing interests:** The authors have declared that no competing interests exist.

11.8 pmp, with rates of 17.4 pmp for males and 6.6 pmp for females, mainly driven by IgAN, rising until 2015 then declining. ADPKD incidence was 9.8 (males) and 8.2 (females) pmp. Incidence varied by age, sex, and deprivation, with higher rates in older adults with MN (IRR 1.75, 95% CI 1.45–2.11) and MPGN (IRR 1.70, 95% CI 1.22–2.38) compared with ADPKD. Pre-emptive transplantation was less common in pGN (13.5%) than ADPKD (22.8%). Overall survival was similar, but outcomes differed by modality: in pGN, 10-year survival was 81.8% after transplantation, 55.5% on PD, and 37.9% on haemodialysis. Among pGN subtypes, IgAN had favourable graft survival and comparable 5-year survival on PD and transplantation (92.1% vs 92.6%). Graft survival was highest in ADPKD.

## Conclusions

This national study reveals evolving epidemiology from ESKD-pGN in the UK. These findings may inform resource allocation, and targeted interventions to enhance care and outcomes in pGN.

## Introduction

Incidence of end stage kidney disease (ESKD) is steadily increasing worldwide, over a period of time [1,2]. Kidney replacement therapy (KRT, dialysis or transplantation) is the mainstay of management for ESKD. Glomerulonephritis is a leading cause worldwide, with primary forms (pGN) including IgA nephropathy, membranous nephropathy, focal segmental glomerulosclerosis (FSGS), and membranoproliferative glomerulonephritis (MPGN). Despite KRT, ESKD is associated with reduced survival and imposes significant lifelong healthcare costs [3].

Although considered rare in general population, pGN as a group contribute to a substantial proportion of patients with ESKD [4]. According to the latest annual report from the UK Renal Registry (UKRR), pGN constitutes the second most common aetiology needing KRT initiation, following diabetes mellitus [4]. 8,175 adult patients started KRT for ESKD in the United Kingdom in 2021, with a rate of 154 per million population (pmp) [5]. This represents an increase of 7.3% compared to figures from 2020 [5]. Despite advances in the understanding of pathobiology and therapeutic interventions for select forms of pGN over the past decade, incidence of KRT attributable to pGN between 2012 and 2021 remains consistent, ranging between 12–14% [5].

The contribution of individual pGN subtypes to ESKD in the UK remains unclear, compared to some other countries [5–7]. In a systematic review of incident GN published in 2011, only one study from the United Kingdom was included [6]. Recent findings from the European registry included a constrained dataset from the United Kingdom [7]. So far, data from the United Kingdom has primarily offered insights into the overall incidence and outcomes of all pGNs combined, and not to individual pGN types. To date, factors related to survival after KRT-pGN are unavailable in the UK.

Understanding incidence and survival is crucial for healthcare services and funding organizations like NICE to appraise the value of new treatments.

This is a descriptive, population-based epidemiological study using national registry data, primarily designed to estimate the incidence of KRT due to primary glomerulonephritis and associated long-term KRT outcomes, including modality use and patient and graft survival, rather than to test causal hypotheses.

## Methods

This observational cohort study enrolled patients with ESKD from kidney centres across the UK via automated data linkage with the UK Renal Registry (UKRR). Eligible participants were adults (≥18 years) initiating KRT between January 1, 1999, and December 31, 2019, with primary diagnoses defined by ERA-EDTA codes (Supplementary material). Date of first KRT was determined from the registry, and anonymised data encompassing treatment and outcome measures were extracted. Full coverage with linkage from all the centres was established by January 1, 2008, and patients were followed up until December 31, 2021. The authors had no access to information that could identify individual participant during or after data collection. The four pGNs (IgAN, FSGS, MN and MPGN) were grouped as 'AnyGN,' and patients with ADPKD were used as the reference disease. ADPKD was included as a reference disease to provide contextual background for incidence and outcome estimates for a common, non–immune-mediated cause of ESKD with stable epidemiology; it was not intended as a matched control or exposure comparator, given its distinct pathogenesis, disease course, and transplantation pathways compared with primary glomerulonephritis. Outcomes were reported for 'AnyGN,' ADPKD, and individual pGN groups, achieving 100% data completeness, except for ethnicity and comorbidities (Supplementary Figure 7 in S1 File). Sensitivity analyses were conducted in subsets with complete comorbidity data to assess the consistency of the direction and magnitude of key associations; these analyses were not powered for formal subgroup comparisons.

Population data were sourced from the Office for National Statistics (ONS) mid-year estimates [8]. The study adhered to the legal basis of section 251 of the NHS Act 2006: 16/CAG/0064, exempting it from separate ethical approval. This report follows the STROBE guidelines for observational research.

### Patient cohorts

We defined two cohorts:

- An incidence cohort consisting of all patients ascertained at first KRT in the period where the registry had full UK coverage (01/01/2008 to 31/12/2019) (Supplementary Figure 4 in S1 File). Only pGN patients initiating KRT were included, so those with conservatively managed pGN (without KRT) were not captured, and incidence reflects KRT initiation rather than overall population pGN

- A survival cohort consisting of all patients ascertained with primary diagnosis and the time of first KRT between 01/01/1999 to 31/12/2019.This extended cohort includes patients diagnosed with KRT where there was not full registry coverage and hence the denominators needed to compute incidence were not available (Supplementary Figure 4 in S1 File).

### Definitions

- Incidence: The date of commencement of KRT and entry into the UKRR (Supplementary Figure 5 in S1 File).

- All cause survival: The differences between dates of first KRT and death, censored at last follow up (Supplementary Figure 5 in S1 File).

- Time to transplant: The difference between first KRT and first transplant dates (Supplementary Figure 6 in S1 File).

- Date of graft failure: The first date where graft failure or another treatment start (transplant or dialysis) was recorded up to and including a subsequent transplant, with graft survival time being censored at the last follow-up date or death (Supplementary Figure 6 in S1 File).

- Delayed transplantation: Transplant after dialysis (Supplementary Figure 6 in S1 File).

## Outcomes and exploratory variables

Baseline characteristics were evaluated at the commencement of KRT. Age was categorised into decades with those aged 18–19 included with the 20–29 age group. Ethnicity was based on patient reported data and categorised into five groups. This was not recorded in substantial proportion and were treated as a separate category in the analyses. Co-morbidity data at initiation of KRT was only available for a proportion of patients and therefore this was not included in the primary survival analyses and a secondary analysis restricted to the subset with available data was performed for co-morbidity covariates.

Deprivation indices were derived by the UKRR based on UK-census derived Index of Multiple Deprivation (IMD) quintiles and the patient's registered residence [5]. Q1 is the most deprived and Q5 the least deprived. As these indices differ between the 4 UK countries and are not available for Scotland, England-only subsets are used to explore differences between IMD quintiles. As the population data is not available by deprivation, we assumed that 1/5 of the relevant population falls within each IMD quintile.

## Statistical methods

Incidence rates are normalised to the relevant age, sex and country population levels for each year. To estimate the effects of demographic factors on incidence of each diagnosis, multivariable Poisson regression models were used with an offset of the logarithm of the relevant population. Preliminary analysis revealed that age and sex were not independent (a substantial age by sex interaction); so, the model fitted included, age-group and sex and their interaction, year (as a categorical variable) and country (England, Scotland, Wales, Northern Ireland). A second model included IMD in an England-only subset. Results are presented as Incidence Rate Ratios (IRR) with 95%CI. Ethnicity was not included due to the large proportion of missing data. No substantial associations regarding missing data were observed related to centre or treatment year. Differential missingness by disease severity or socioeconomic factors cannot be excluded and its potential impact, including for ethnicity, should be considered when interpreting subgroup estimates.

Analysis of overall survival utilised Cox regression models with a time-dependent covariate for KRT during each time period for each patient whilst on each treatment modality. Preliminary analyses revealed strong non-proportionality; therefore, stratified models were fitted with separate baseline hazards for all combinations of diagnosis, KRT and age group:

- The primary model (Model 1) adjusted for age group, diagnosis, KRT and covariates for sex and ethnicity.

- Model 2: included IMD quintile in an England-only subset in addition to model 1 adjustments.

- Model 3: included adjustment for comorbidities and smoking in the subset where these were available in addition to model 1 and 2 adjustments.

Results are presented as fitted survival rates and 1, 5, and 10 years with 95% CI. Graft survival was assessed using the same models as overall survival. Consistent with contemporary epidemiological reporting standards, analyses was focussed on estimation of effect sizes with corresponding confidence intervals rather than hypothesis testing based on P-values. Comparisons are descriptive and hypothesis-generating, and no formal adjustment for multiple comparisons was undertaken.

 

## Results

### Patient characteristics

10,308 and 8,302 patients were in pGN and ADPKD survival cohorts respectively. From these cohorts, 7,206 with pGN and 5,471 with ADPKD were ascertained in the incidence cohorts, as outlined in Table 1. Median age at initiation of KRT due to pGNs was 54 years (IQR: 41–67), patients with MN at first KRT were older at 67 years (IQR: 56–75) compared to 50 years (IQR: 38–62) with IgAN. Most patients were white (89.1% in ADPKD and 78.7% in pGN). In the pGN group, a male predominance was observed at 71.6% compared to 53.1% in ADPKD (Table 1). Comorbidity data was available for 64.9% of patients in incident cohort and outlined in Table S2a and S2b in S1 File.

Haemodialysis (HD) was the most common initial modality for KRT (59.6% in pGN vs 51.4% in ADPKD). Pre-emptive transplantation was more frequently seen in the ADPKD group (22.8%) compared to pGN (13.5%) (Table 1). Across the types of pGN, patients with MN had the lowest pre-emptive transplant incidence of 6.3% (Table 1). For those with delayed transplantation it was shorter in ADPKD group (median 265 days) compared to pGN (median 383 days) with MN patients

**Table 1.  Incident Cohort Demographics.**

| | | ADPKD | pGN | FSGS | IgAN | MN | MPGN |
|---|---|---|---|---|---|---|---|
| | N | 5471 | 7206 | 1391 | 4147 | 1082 | 586 |
| Age First KRT | Median [IQR] | 55 (48-64) | 54 (41-67) | 56 (43-68) | 50 (38-62) | 67 (56-75) | 58 (43-70) |
| | Range | [21-95] | [18-93] | [18-90] | [18-93] | [19-91] | [18-89] |
| | | | | | | | |
| Sex | Male | 2906 (53.1%) | 5159 (71.6%) | 867 (62.3%) | 3155 (76.1%) | 780 (72.1%) | 357 (60.9%) |
| | | | | | | | |
| Ethnicity | White | 4453 (89.1%) | 5158 (78.7%) | 905 (71.4%) | 3000 (79.4%) | 801 (81.7%) | 452 (85.8%) |
| | Asian | 259 (5.2%) | 849 (12.9%) | 170 (13.4%) | 534 (14.1%) | 101 (10.3%) | 44 (8.3%) |
| | Black | 183 (3.7%) | 341 (5.2%) | 146 (11.5%) | 115 (3%) | 57 (5.8%) | 23 (4.4%) |
| | Mixed | 47 (0.9%) | 75 (1.1%) | 21 (1.7%) | 45 (1.2%) | 7 (0.7%) | 2 (0.4%) |
| | Other | 57 (1.1%) | 133 (2%) | 26 (2.1%) | 86 (2.3%) | 15 (1.5%) | 6 (1.1%) |
| | Missing | {472 (9%)} | {650 (9%)} | {123 (9%)} | {367 (9%)} | {101 (9%)} | {59 (10%)} |
| Country | England | 4455 (81.4%) | 5837 (81%) | 1153 (82.9%) | 3357 (81%) | 859 (79.4%) | 468 (79.9%) |
| | Scotland | 532 (9.7%) | 744 (10.3%) | 129 (9.3%) | 439 (10.6%) | 118 (10.9%) | 58 (9.9%) |
| | Wales | 295 (5.4%) | 439 (6.1%) | 92 (6.6%) | 232 (5.6%) | 70 (6.5%) | 45 (7.7%) |
| | N Ireland | 189 (3.5%) | 186 (2.6%) | 17 (1.2%) | 119 (2.9%) | 35 (3.2%) | 15 (2.6%) |
| | | | | | | | |
| IMD Quintile | N | 4455 | 5837 | 1153 | 3357 | 859 | 468 |
| (England Only) | Q1 | 858 (19.3%) | 1326 (22.7%) | 269 (23.3%) | 774 (23.1%) | 176 (20.5%) | 107 (22.9%) |
| | Q2 | 914 (20.5%) | 1272 (21.8%) | 257 (22.3%) | 729 (21.7%) | 173 (20.1%) | 113 (24.1%) |
| | Q3 | 948 (21.3%) | 1213 (20.8%) | 239 (20.7%) | 675 (20.1%) | 193 (22.5%) | 106 (22.6%) |
| | Q4 | 879 (19.7%) | 1091 (18.7%) | 207 (18%) | 633 (18.9%) | 180 (21%) | 71 (15.2%) |
| | Q5 | 856 (19.2%) | 935 (16%) | 181 (15.7%) | 546 (16.3%) | 137 (15.9%) | 71 (15.2%) |
| | | | | | | | |
| Initial Treatment | Transplant | 1245 (22.8%) | 972 (13.5%) | 159 (11.4%) | 675 (16.3%) | 68 (6.3%) | 70 (11.9%) |
| | HD | 2811 (51.4%) | 4292 (59.6%) | 892 (64.1%) | 2220 (53.5%) | 798 (73.8%) | 382 (65.2%) |
| | PD | 1415 (25.9%) | 1942 (26.9%) | 340 (24.4%) | 1252 (30.2%) | 216 (20%) | 134 (22.9%) |

Baseline demographics of the incident cohort with APKD and the four primary glomerular nephropathies: IgAN, FSGS, MN, and MPGN.

taking the longest (623 days) after dialysis initiation (Supplementary Table 5 in S1 File). During follow up, one-third of patients did not receive a transplant in ADPKD group, compared to 41.5% in pGN group.

**Incidence of KRT**

Fig 1 and Table 2 illustrate the crude incidence rates of KRT. Over the study period, the incidence of KRT due to pGN was 11.8 per million population, with incidence of 17.4 per million in males and 6.6 pmp in females (Table 2). Progressive increase in KRT incidence in pGN compared to the ADPKD groups was seen, especially between 2008 and 2016 with gradual reduction in the more recent years (Fig 1A). There was substantial difference in KRT incidence between males and females in AnyGN, with a profound difference in IgAN, where male to female ratio was nearly 3:1 (Tables 1 and 2). KRT incidence increases with age, which is more pronounced in IgAN and in MN during later years (Fig 2, Table S4 in S1 File). There was substantial and progressive reduction in the incidence of KRT-pGN in areas with least deprivation (Fig 2C, F, Table S3 in S1 File).

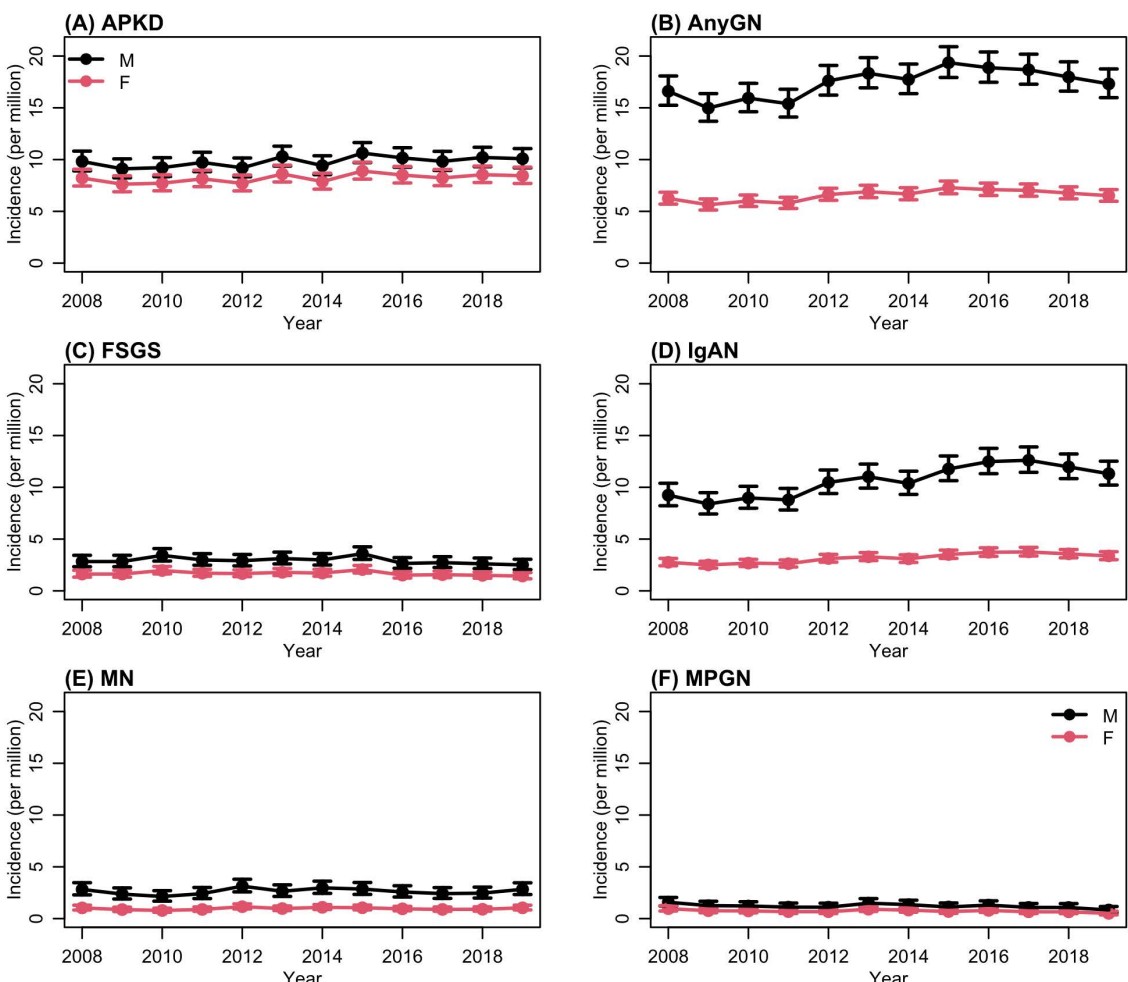

**Fig 1. Overall Incidence per million of each type by sex over time.** Estimates from a Poisson regression model with denominators taken from the UK population.

**Table 2. Overall KRT incidence by primary disease.**

|  | Male | Female |
|---|---|---|
| **AnyGN** | 17.4 (17.0-17.9) | 6.6 (6.3-6.8) |
| FSGS | 2.9 (2.7-3.1) | 1.7 (1.5-1.8) |
| IgAN | 10.7 (10.3-11.0) | 3.2 (3.0-3.4) |
| MN | 2.6 (2.5-2.8) | 1.0 (0.9-1.1) |
| MPGN | 2.6 (2.5-2.8) | 1.0 (0.9-1.1) |
| **ADPKD** | 9.8 (9.5-10.2) | 8.2 (7.9-8.5) |

Overall KRT incidence rates per million population with 95%CI for pGN, the 4 pGN subtypes and ADPKD.

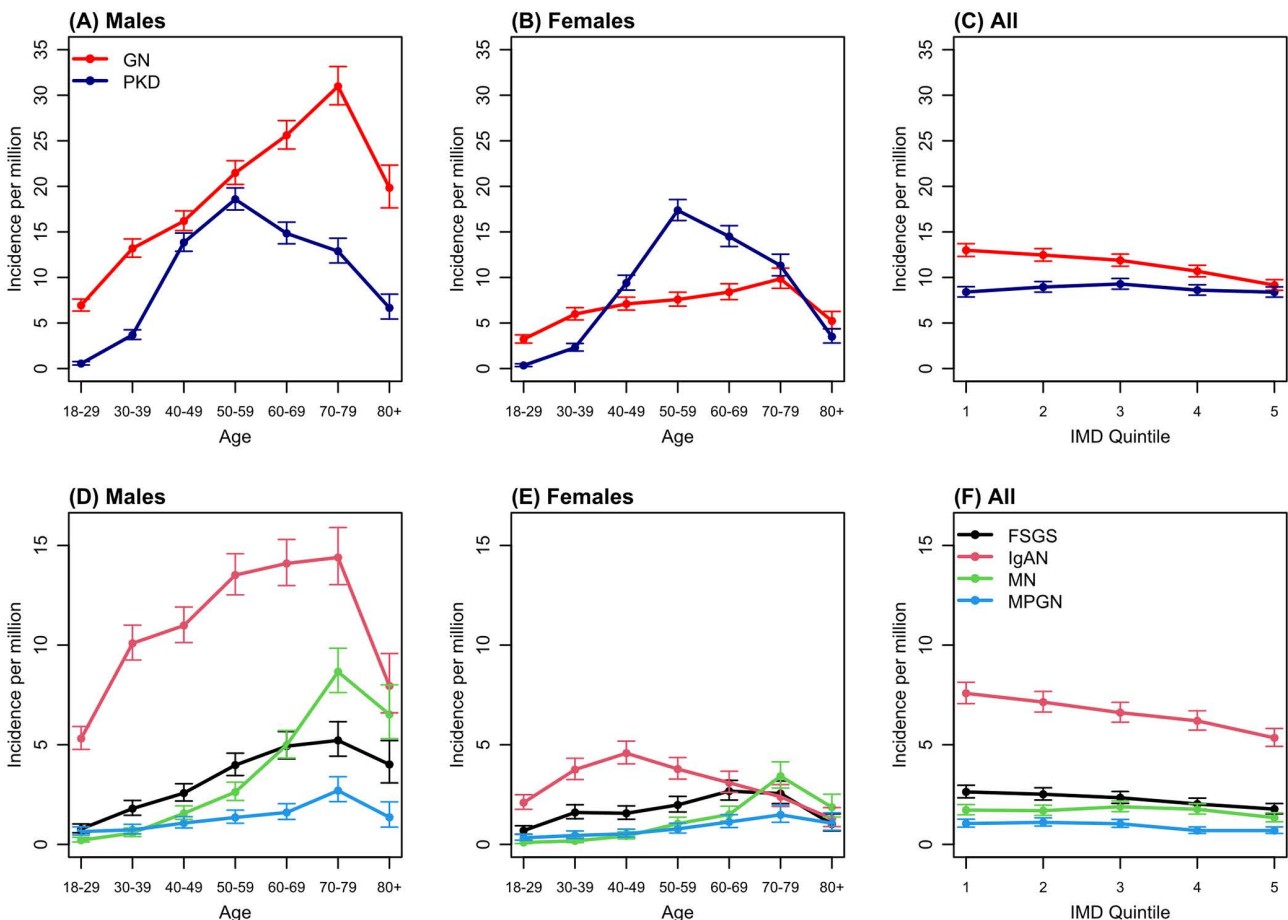

**Fig 2. Incidence per million population for pGN compared to APKD (A-C) and the 4 pGN subtypes (D-E).** Panels (A), (B) (D) and (E) show the incidence by age for males and females. Panels (C) and (F) show the English incidence by deprivation as measured by the IMD quintiles (1 being the least and 5 the most deprived). Estimates are derived from a Poisson regression model adjusting for year, age, sex, country and (panels (C) and (F)) IMD.

## Patient survival after KRT

Table S1 in S1 File outlines the baseline characteristics and demographics of patients in survival cohort. Comorbidity data recorded at the time initiation of KRT in the survival cohort was available in 11,558 patients (62.1% of patients with pGN and ADPKD combined). At least one of the listed comorbidities was present in 34.2% in pGN group and 30.1% in ADPKD group. The most common comorbidity was cardiovascular disease in 17% of patients with both pGN and ADPKD. MN carries the highest burden of cardiovascular diseases among pGNs (Table S2a and b in S1 File).

## Patient survival

Figs 3–4 and Table 4 illustrate the covariate-adjusted overall survival for pGNs and ADPKD cohorts following KRT initiation. Unadjusted survival curves are shown in Supplementary Figure S1 in S1 File. The overall survival between ADPKD and pGN was comparable throughout the years. This was similar when comparing the four types of pGN collectively, with no significant difference in survival observed over the years from the initiation of KRT (Fig 3). There is a wide disparity in survival between each KRT modality (HD, PD and transplant) across all the disease categories in favour for transplantation (Fig 4). Survival rates with dialysis is significantly less compared to those with transplantation (Table 3). Younger patients aged 18–29 generally exhibit excellent survival outcomes across all disease groups (Table S6 in S1 File).

## Transplant graft survival

Overall graft survival (death-censored) following KRT initiation demonstrated superiority in individuals with ADPKD compared to those with pGN (Fig 5). Unadjusted survival curves are shown in Supplementary Figure 2 in S1 File. The adjustments for socioeconomic status, comorbidities, and smoking have relatively minor effects on overall graft survival (Table 5). Among the four pGNs, patients with IgA nephropathy had better graft survival following KRT compared to other pGNs (Fig 5). Graft survival was notably favourable following pre-emptive transplantation compared to transplantation after dialysis initiation (Supplementary figure 3 in S1 File).

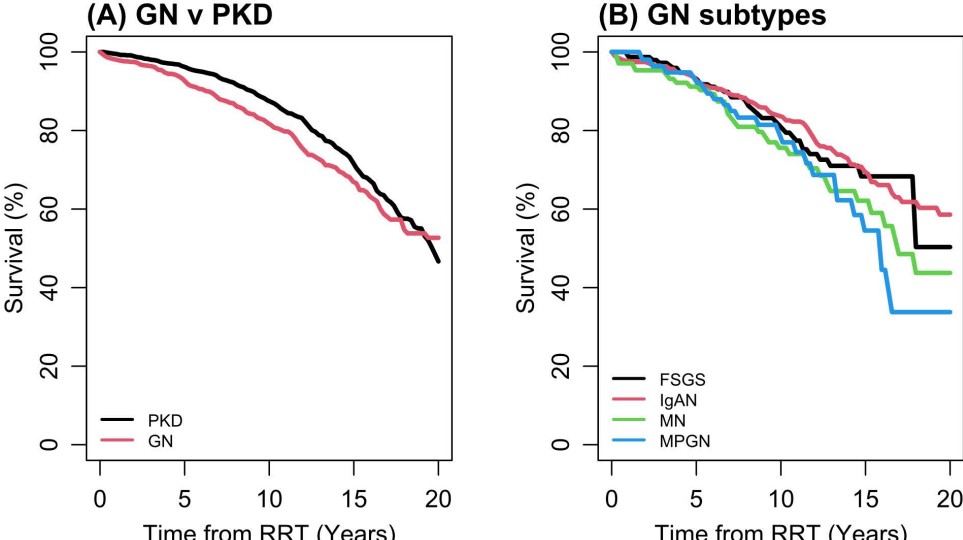

**Fig 3. Fitted overall survival from the stratified Cox model.** These adjusted estimates are referenced to male, age 60-69, white with KT as treatment. Based on primary model [1]. Unadjusted Kaplan-Meier survival curves including numbers at risk are provided in Supplementary figure 1 in S1 File.

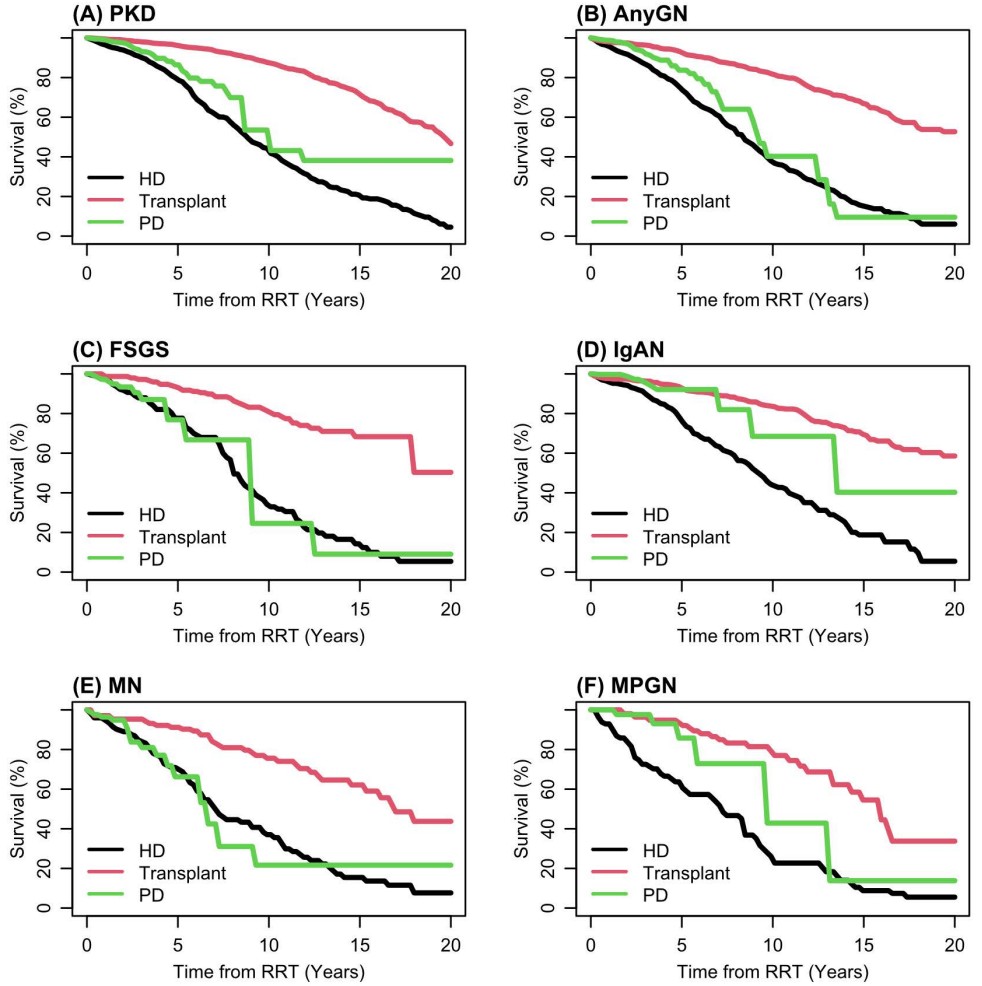

**Fig 4. Fitted overall survival from the stratified Cox model for each KRT modality.** These adjusted estimates are referenced to male, age 60-69, white ethnicity. Based on the primary model [1].

## Discussion

This is one of the larger registry studies in literature. Rates of pGN-ESKD were previously reported in registry studies outside the UK. In the ANZDATA registry spanning 1998–2012, IgAN-ESKD incidence was reported at 6 per million population (pmp) [9]. More recent European data reveals an overall incidence of 16.6 pmp for all pGNs. Particularly noteworthy is the an upward trajectory observed for IgA nephropathy (IgAN) at 4.6 pmp, and focal segmental glomerulosclerosis FSGS at 2.6 pmp, in comparison to lower rates seen in MN at 0.7 pmp, and MPGN at 1.4 pmp [7]. Our results, with overall incidence of 11.8 pmp for pGN, are consistent with these findings. In this national cohort study over 20 years, we found that KRT incidence was higher and pre-emptive transplantation rates were lower in pGN than those with ADPKD. Whilst patient survival rates were comparable, graft survival was poorer in pGN compared to ADPKD. Pre-emptive transplantation was associated with better graft survival.

Among the subtypes, IgA nephropathy has the highest incidence at 6.95 pmp, followed by FSGS at 2.3 pmp, while MN and MPGN have the lowest incidence, each at 1.8 pmp. Conversely, studies from the United States consistently showed that FSGS was the largest contributor to ESKD, [10]. In the largest study published on pGN-ESKD during 1996–2011,

**Table 3. Overall patient survival rates by KRT.**

| Group | KRT | 1 Years | 5 Years | 10 Years |
|---|---|---|---|---|
| ADPKD | Transplant | 99.5 (98.9-100.0) | 96.2 (95.1-97.3) | 87.6 (85.7-89.5) |
| | HD | 96.5 (95.5- 97.5) | 79.0 (76.1-82.0) | 43.2 (38.5-48.5) |
| | PD | 99.0 (98.3- 99.7) | 86.4 (81.8-91.3) | 53.5 (39.4-72.8) |
| pGN | Transplant | 98.0 (96.5-99.5) | 92.4 (90.4-94.5) | 81.8 (78.9-84.7) |
| | HD | 95.7 (94.5-96.9) | 74.4 (71.2-77.7) | 37.9 (33.5-43.0) |
| | PD | 98.8 (97.9-99.7) | 83.7 (78.1-89.7) | 40.3 (23.9-67.9) |
| FSGS | Transplant | 98.7 (96.2-100.0) | 93.0 (88.7-97.4) | 80.4 (73.9- 87.5) |
| | HD | 97.7 (95.6- 99.7) | 77.7 (71.4-84.5) | 34.1 (25.7- 45.4) |
| | PD | 97.2 (94.2-100.0) | 76.9 (62.5-94.5) | 24.5 (3.4-100.0) |
| IgAN | Transplant | 97.7 (95.7- 99.7) | 92.6 (90.0-95.2) | 83.7 (80.3- 87.3) |
| | HD | 95.8 (94.2- 97.5) | 77.0 (72.5-81.8) | 44.2 (37.5- 52.2) |
| | PD | 99.7 (99.2-100.0) | 92.1 (87.3-97.3) | 68.5 (44.8-100.0) |
| MN | Transplant | 97.1 (91.6-100.0) | 91.2 (84.1-98.9) | 75.6 (66.5-86.0) |
| | HD | 94.7 (91.4- 98.2) | 69.9 (62.4-78.4) | 37.0 (28.3-48.5) |
| | PD | 96.4 (92.4-100.0) | 66.2 (50.5-86.8) | 21.6 (6.8-68.7) |
| MPGN | Transplant | 100.0 (100.0-100.0) | 92.1 (85.7- 99.1) | 79.2 (69.6- 90.2) |
| | HD | 92.9 (88.3- 97.8) | 62.1 (52.7- 73.3) | 26.1 (17.1- 39.7) |
| | PD | 100.0 (100.0-100.0) | 85.8 (70.9-100.0) | 42.8 (14.2-100.0) |

1-, 5- and 10-year adjusted survival rates for each group with 95%CI alongside the number of treatment episodes and patients. These estimates are adjusting for age, sex, ethnicity and KRT and referenced to male, age 60–69, white.

34,330 out of 84,301 (40.7%) patients with pGN had FSGS as primary disease [10]. In contrast to our findings, a Japanese study reported a notable downward trend in KRT incidence for IgAN over the past three decades. Such observations underscore the ethnic heterogeneity in IgAN, and a complex interplay of various factors including decisions on interventions [11].

Incidence of KRT in ADPKD was consistent across various IMD quintiles in contrast to pGNs where this was worse, particularly in the least deprived areas irrespective of the type of pGN. Our findings regarding IMD in England closely parallel those observed in the Scottish registry, revealing a comparable decline in KRT incidence within the least deprived regions [12]. Further comprehensive research is warranted to elucidate the underlying contributing factors behind this observation. We observed no difference in overall survival between ADPKD and pGNs after KRT, nor among the four pGN subtypes. In other studies, MN is associated with the highest mortality rate among the four pGNs [13,14].

Existing literature consistently demonstrates that transplantation provides superior survival outcomes compared to dialysis [15]. Our study observed that Peritoneal dialysis was associated with survival outcomes comparable to transplantation in selected subgroups, particularly IgAN. These findings are purely observational and subject to treatment selection bias or residual confounding; they should not be interpreted as evidence to guide therapy, but rather as hypotheses that require confirmation in prospective studies. Similar patterns have been reported in other populations, where the early survival advantage of PD diminishes over time [16]. Lack of this association with other pGNs and with ADPKD highlights the need for further research to determine the optimal dialysis modality in ESKD-pGN. Regardless of the KRT modality, we observed that survival across all disease categories is influenced by factors such as smoking, comorbidities, age, and socioeconomic status (IMD). Given limited comorbidity data, adjustment for confounding was restricted, and unmeasured factors such as disease severity and selection by indication may have influenced survival differences; however,

**Table 4. Adjusted patient survival rates.**

| model | group | Episodes | Patients | 1 Year | 5 Years | 10 Years |
|-------|-------|----------|----------|--------|---------|----------|
| Model 1 | ADPKD | 15,716 | 8,302 | 99.5 (98.9-100.0) | 96.2 (95.1-97.3) | 87.6 (85.7-89.5) |
| | pGN | 20,547 | 10,308 | 98.0 (96.5- 99.5) | 92.4 (90.4-94.5) | 81.8 (78.9-84.7) |
| | FSGS | 3,849 | 1,934 | 98.7 (96.2-100.0) | 93.0 (88.7-97.4) | 80.4 (73.9-87.5) |
| | IgAN | 11,909 | 5,724 | 97.7 (95.7- 99.7) | 92.6 (90.0-95.2) | 83.7 (80.3-87.3) |
| | MN | 2,775 | 1,636 | 97.1 (91.6-100.0) | 91.2 (84.1-98.9) | 75.6 (66.5-86.0) |
| | MPGN | 2,014 | 1,014 | 100.0 (100.0-100.0) | 92.1 (85.7-99.1) | 79.2 (69.6-90.2) |
| Model 2 | ADPKD | 12,547 | 6,621 | 99.3 (98.6-100.0) | 96.0 (94.7-97.3) | 86.4 (84.1-88.8) |
| | pGN | 16,410 | 8,202 | 97.4 (95.4- 99.3) | 91.0 (88.4-93.7) | 80.4 (76.9-84.1) |
| | FSGS | 3,153 | 1,583 | 98.6 (95.8-100.0) | 93.1 (88.4-98.0) | 79.8 (72.3-88.0) |
| | IgAN | 9,436 | 4,536 | 97.0 (94.4- 99.7) | 90.9 (87.6-94.4) | 82.2 (77.9-86.8) |
| | MN | 2,164 | 1,256 | 95.7 (87.9-100.0) | 87.9 (78.3-98.6) | 75.0 (64.1-87.7) |
| | MPGN | 1,657 | 827 | 100.0 (100.0-100.0) | 92.1 (84.9-99.9) | 77.6 (66.6-90.6) |
| Model 3 | ADPKD | 10,039 | 5,197 | 99.6 (99.0-100.0) | 97.7 (96.6-98.8) | 90.0 (87.9-92.1) |
| | pGN | 13,333 | 6,361 | 99.5 (98.5-100.0) | 94.0 (91.9-96.2) | 85.0 (81.9-88.3) |
| | FSGS | 2,488 | 1,180 | 100.0 (100.0-100.0) | 96.3 (92.7- 99.9) | 86.5 (79.9- 93.8) |
| | IgAN | 7,798 | 3,596 | 99.3 (98.0-100.0) | 93.4 (90.7- 96.2) | 85.9 (82.0- 89.9) |
| | MN | 1,760 | 981 | 100.0 (100.0-100.0) | 93.2 (87.0- 99.9) | 78.3 (68.5- 89.6) |
| | MPGN | 1,287 | 604 | 100.0 (100.0-100.0) | 97.9 (93.8-100.0) | 90.0 (81.0-100.0) |

1-, 5- and 10-year adjusted survival rates for each group with 95%CI alongside the number of treatment episodes and patients. Estimates from Cox regression models. Model 1 fits the full survival dataset adjusting for age, sex, ethnicity and treatment. Model 2 additionally adjusts for deprivation in the English subset and model 3 adjusts for smoking and comorbidities in the subset where these are available. These estimates are referenced to male, age 60–69, white.

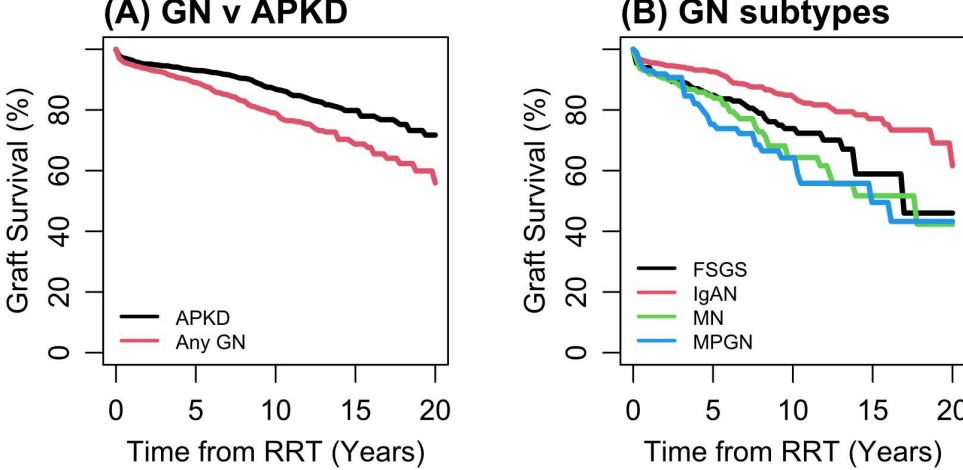

**Fig 5. Fitted graft survival from the stratified Cox model.** These adjusted estimates are referenced to male, age 60-69, white. Based on primary model [1]. Unadjusted Kaplan-Meier survival curves including numbers at risk are provided in Supplementary figure 2 in S1 File.

**Table 5. Graft survival estimates.**

| model | group | Patients | 1 Year | 5 Years | 10 Years |
|---|---|---|---|---|---|
| Model 1 | ADPKD | 5,390 | 96.4 (95.6-97.2) | 93.1 (91.9-94.3) | 86.8 (84.9-88.8) |
| | pGN | 5,969 | 94.9 (93.7-96.1) | 89.1 (87.3-90.9) | 78.6 (75.7-81.7) |
| | FSGS | 1,058 | 93.9 (91.1-96.9) | 84.8 (80.3-89.7) | 73.9 (67.1-81.3) |
| | IgAN | 3,893 | 95.8 (94.4-97.2) | 92.6 (90.7-94.5) | 84.0 (80.7-87.6) |
| | MN | 546 | 91.9 (87.7-96.2) | 84.9 (79.2-91.1) | 64.3 (54.6-75.7) |
| | MPGN | 472 | 92.9 (88.0-98.1) | 75.3 (66.4-85.3) | 64.2 (53.6-76.9) |
| Model 2 | ADPKD | 4,312 | 96.7 (95.8-97.6) | 93.9 (92.6-95.3) | 88.0 (85.7-90.3) |
| | pGN | 4,771 | 95.5 (94.3-96.8) | 89.5 (87.4-91.7) | 78.9 (75.3-82.7) |
| | FSGS | 870 | 94.2 (91.2-97.3) | 84.2 (79.0-89.7) | 73.5 (65.9-82.0) |
| | IgAN | 3,089 | 95.9 (94.4-97.5) | 92.6 (90.3-94.9) | 83.5 (79.3-87.9) |
| | MN | 433 | 93.2 (88.9-97.7) | 86.6 (80.5-93.3) | 65.5 (54.6-78.6) |
| | MPGN | 379 | 94.2 (89.4-99.3) | 75.1 (65.3-86.4) | 61.5 (49.3-76.7) |
| Model 3 | ADPKD | 3,460 | 96.5 (95.5-97.5) | 93.7 (92.4-95.1) | 87.4 (85.1-89.8) |
| | pGN | 3,838 | 95.4 (94.1-96.8) | 89.3 (87.1-91.6) | 78.0 (74.3-81.9) |
| | FSGS | 685 | 92.8 (89.1-96.7) | 84.3 (78.7-90.3) | 72.8 (64.7-82.0) |
| | IgAN | 2,501 | 96.8 (95.3-98.3) | 93.4 (91.1-95.7) | 83.5 (79.2-88.0) |
| | MN | 356 | 92.6 (87.7-97.7) | 86.9 (80.3-94.0) | 66.8 (54.6-81.6) |
| | MPGN | 296 | 93.9 (88.3-99.9) | 68.9 (56.9-83.6) | 57.7 (44.3-75.1) |

Fitted graft survival adjusted using the 3 models.

Graft survival estimates. Model 1 fits the full survival dataset adjusting for age, sex, ethnicity and treatment. Model 2 additionally adjusts for deprivation in the English subset and model 3 adjusts for smoking and comorbidities in the subset where these are available. These estimates are referenced to male, age 60–69, white.

the direction and magnitude of key associations remained consistent in sensitivity analyses restricted to patients with complete comorbidity data, supporting the robustness of the main findings despite limited power for formal subgroup conclusions.

Rates of pre-emptive transplantation in pGNs are substantially different to patients affected by ADPKD. This disparity in incidence may be attributed to variability in age presentation between MN and other pGNs, which may potentially be a barrier to pre-emptive transplantation in general. However, it is of interest that the pre-emptive transplantation rate is higher in this study at 13.5% in all pGNs combined together compared to rates of 6.8% and 7.2% in the European and USRDS studies respectively [7,10]. Pre-emptive transplantation demonstrated superior survival outcomes compared to delayed transplantation following dialysis initiation [17]. Other studies have indicated that pre-emptive transplantation is less common among racial minorities and individuals with lower levels of education [18].

Our findings show graft survival rates for pGNs are lesser than those with ADPKD, with IgAN demonstrating better outcomes than other pGNs. This may be due to IgAN's earlier onset, allowing better-matched transplants via the UK allocation scheme. Additionally, higher rates of pre-emptive transplants for ADPKD may explain its superior outcomes. Efforts should focus on earlier transplant referrals to improve pre-emptive transplant rates. Registry studies also confirm poorer graft outcomes in pGNs, partly due to recurrent disease, particularly in MPGN [18], leading to higher rates of second transplants in pGN patients compared to ADPKD.

## Limitations

Several limitations inherent to registry-based observational studies should be considered when interpreting these findings. First, although the UK Renal Registry collects data prospectively, the present analyses were conducted retrospectively.

Second, comorbidity data were available for approximately two-thirds of the cohort, and disease severity measures were not captured whilst there was missing data with ethnicity, limiting the ability to fully adjust for clinical risk. Third, primary renal diagnoses were based on registry coding (ERA–EDTA classifications) rather than central pathology review, and reasons for graft failure were unavailable. In addition, the MPGN category did not distinguish between primary complement-mediated disease and secondary causes such as infection or monoclonal gammopathy, which may introduce heterogeneity within this subgroup.

No formal adjustment for multiple comparisons was applied, as the analyses were descriptive and hypothesis-generating in nature. Residual confounding remains possible, including confounding by indication in dialysis modality selection and timing of transplantation. Although sensitivity analyses incorporating comorbidity data demonstrated consistent direction and magnitude of associations, incomplete covariate data and absence of detailed disease severity metrics limit causal interpretation. Accordingly, observed differences in survival should be interpreted as associations rather than evidence of treatment effect.

Furthermore, the registry captures only patients initiating kidney replacement therapy; individuals with pGN managed conservatively or with delayed progression are not included. Therefore, the reported incidence reflects KRT-incidence due to pGN rather than overall population incidence of disease. Despite these limitations, strengths of the study include nationwide coverage, high data completeness, minimal loss to follow-up, and the inclusion of ADPKD as a reference disease to contextualise incidence and outcome patterns.

## Conclusions

This study underscores the significant healthcare burden posed by pGN in the United Kingdom, highlighting its rising incidence and the critical need for improved management strategies. Incidence rates are associated with individual pGN subtypes, age, sex, and socioeconomic factors. The superior survival outcomes associated with pre-emptive transplantation and peritoneal dialysis in specific pGN subtypes warrant further exploration to optimise treatment protocols. These insights are vital for healthcare providers and policymakers to enhance patient care and allocate resources effectively.

## Supporting information

**S1 File. Supporting information including ERA-EDTA codes, Supplementary Tables S1, S2a, S2b, S3, S4a–S4d, S5, S6, S7, and Supplementary Figures S1–S7.**
(DOCX)

## Acknowledgments

We would like to thank all the kidney centres for their contribution of the data to the UKRR, and all the staff at UK Renal Registry. We acknowledge support of Retha Steenkamp, Tom Gray at the UKRR for their support with data access and collaborators in the GN-UNITY study group in the UK.

## Author contributions

**Conceptualization:** Stephen A Roberts, Omar Ragy, Durga AK Kanigicherla.

**Data curation:** Stephen A Roberts, Seb Bate.

**Formal analysis:** Stephen A Roberts, Omar Ragy, Durga AK Kanigicherla.

**Methodology:** Stephen A Roberts.

**Resources:** Anna Casula.

**Software:** Stephen A Roberts.

**Writing – original draft:** Omar Ragy, Durga AK Kanigicherla.

**Writing – review & editing:** Stephen A Roberts, Omar Ragy, Anna Casula, Sandip Mitra, Jonathan Barratt, Anirudh Rao, Durga AK Kanigicherla.

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
