## [Decision Letter · Decision Letter 0]

2 Nov 2025

PONE-D-25-50346

ESKD from Primary Glomerulonephritis: Incidence and post KRT survival: A national cohort study in the UK

PLOS ONE

Dear Dr. Ragy,

Thank you for submitting your manuscript to PLOS ONE. After careful consideration, we have decided that your manuscript does not meet our criteria for publication and must therefore be rejected.

Specifically:

I am sorry that we cannot be more positive on this occasion, but hope that you appreciate the reasons for this decision.

Kind regards,

Mohamed E Elrggal

Academic Editor

PLOS ONE

Additional Editor Comments:

We thank the author for their submission, however, there are major limitations to the concept of the paper highlighted by the reviewers, hindering the further progress of the manuscript. The reviewers comments are highlighted to assist you in improving the quality of the manuscript.

Reviewers' comments:

Reviewer's Responses to Questions

**Comments to the Author**

1. Is the manuscript technically sound, and do the data support the conclusions?

Reviewer #1: Yes

Reviewer #2: Partly

2. Has the statistical analysis been performed appropriately and rigorously?

Reviewer #1: Yes

Reviewer #2: No

3. Have the authors made all data underlying the findings in their manuscript fully available?

Reviewer #1: Yes

Reviewer #2: Yes

4. Is the manuscript presented in an intelligible fashion and written in standard English?

Reviewer #1: Yes

Reviewer #2: Yes

Reviewer #1: The article is overall very informative and well written.

Some comments still are necessary to address:

The title does not reflect the methodology, as it does not comment on the fact that APKD was a comparator. Why was it chosen as a comparator? This is not explained in the manuscript.

It would be better to refer to APKD with the more formal name of “autosomal dominant polycystic kidney disease (ADPKD)”

The language needs to be revised in several sections.

Introduction:

“Incidence of end stage kidney disease (ESKD) is steadily increasing worldwide, over a period of time (1, 2).” Over what period of time?

“Although considered rare in general population,” should be “in the general population”

“pGN as a group contribute” should be “contributes”

Methods:

The retrospective nature of the study is not pointed out in the methods.

“This was not recorded in substantial proportion, and were treated as a separate category in the analyses.” Should be “in a substantial proportion” and “(this variable/ethnicity) was”

“fitted survival rates and 1, 5, and 10 years” should be “fitted survival rates at 1, 5, and 10 years”

Results:

As the main aim is the incidence, it should be reported first.

The tables and supplemental material were not included in the provided manuscript file.

Discussion:

The link between migration and their impact on ethnicity in the UK therefore changing incidence of different types of pGN is not mentioned. Another factor that may have affected incidence and survival is the aging of populations.

Reviewer #2: ESKD from Primary Glomerulonephritis: Incidence and post KRT survival: A national cohort study in the UK

General comments

1. Thanks to the authors for the work done and the interest to share with the scientific world

2. The topic biases readers to think there is discussion of primary GNs and not sure why largely authors are rather comparing to ADPKD? Was rather thinking authors can look at comparing their major causes of GNs

3. Most of the comparisons are not done analytically. Looks mainly descriptive and it might help to know of these differences are statistically significant?

Abstract

1. Authors report regression models in the methods but just reports description in the results

Introduction

1. Authors report IgA nephropathy (IgAN), membranous nephropathy (MN), focal segmental glomerulosclerosis (FSGS), and membranoproliferative glomerulonephritis (MPGN) as major causes of GN in the cohort and was rather expecting anylsis of these and not ADPKD?

2. Might help not to start sentences with Arabic numerals.

Methods

1. Authors should report with some short notes on the UK renal registry for the benefit of the readership

2. Line 138 - Incidence rates are normalised to the relevant age, sex and country” – not sure if authors meant standardized?

3. Line 159 - “Results are presented as fitted survival rates and 1, 5, and 10 years with 95% CI” – did not appreciate a lot of the 95% confidence interval the results

Results

1. Line 169-170 “Most patients were white (89.1% in APKD and 78.7% in pGN). In the pGN

group, a significant male predominance was observed at 71.6% compared to 53.1% in APKD” – was this proven to be statistically significant?

2. Authors should hope to report the significant of all the analysis statistically

discussion

1. Might be helpful to have looked at the GNs as per topic and not sure why authors and would have loved to see how the various GNs progress and their transplant outcomes.

.

Reviewer #1: **Yes:** Yasmine NagaYasmine NagaYasmine NagaYasmine Naga

Reviewer #2: No

- - - - -

---

## [Author Response · Author response to Decision Letter 1]

16 Dec 2025

Incidence of ESKD and long-term outcomes in patients with primary glomerulonephritis - A UK National Cohort Study

PONE-D-25-50346

Reviewer's Responses to Questions

We thank the editorial team and both reviewers for their thoughtful assessments. Where suggestions were helpful, we have incorporated changes. Where concerns were raised or based on misunderstandings, we would clarify the text and provide explanation.

Comments to the Author

1. Is the manuscript technically sound, and do the data support the conclusions?

Reviewer #1: Yes

Reviewer #2: Partly

We respectfully note that Reviewer 2 has not presented any evidence to support this assessment

2. Has the statistical analysis been performed appropriately and rigorously?

Reviewer #1: Yes

Reviewer #2: No

We strongly dispute this assessment: There are no examples presented where the analysis is questionable. Rather the reviewer seems to be looking for the results focused on P-values rather than effect sizes and their confidence intervals (this is a rather outdated presentation, as has been recommended since the seminal paper of Gardner in the BMJ in 1986 and picked up in recent years in, for example, a series of papers in Nature). The analysis and presentation follow standard practice in epidemiological studies where the aim is to describe the various populations in a comparable manner.

We also note that following the reproducible research paradigm, the statistical analysis was pre-specified in the Protocol and in a Statistical Analysis Plan which were both reviewed and agreed prior to the analysis.

3. Have the authors made all data underlying the findings in their manuscript fully available?

The PLOS Data policy [track.editorialmanager.com] requires authors to make all data underlying the findings described in their manuscript fully available without restriction, with rare exception (please refer to the Data Availability Statement in the manuscript PDF file). The data should be provided as part of the manuscript or its supporting information or deposited to a public repository. For example, in addition to summary statistics, the data points behind means, medians and variance measures should be available. If there are restrictions on publicly sharing data—e.g. participant privacy or use of data from a third party—those must be specified.

Reviewer #1: Yes

Reviewer #2: Yes

4. Is the manuscript presented in an intelligible fashion and written in standard English?

Reviewer #1: Yes

Reviewer #2: Yes

5. Review Comments to the Author

Reviewer #1

The article is overall very informative and well written.

Some comments still are necessary to address:

1. The title does not reflect the methodology, as it does not comment on the fact that APKD was a comparator. Why was it chosen as a comparator? This is not explained in the manuscript.

We thank the reviewer for this helpful comment. We have revised the Introduction to clearly explain the rationale for including ADPKD as a comparator. ADPKD was selected because it is a common, non-immune mediated disease, well-characterised cause of ESKD with stable epidemiology, predictable natural history, and highly reliable coding within the UKRR. These features make ADPKD an ideal benchmark condition against which rarer pGNs can be meaningfully compared. In epidemiological research, crude incidence estimates alone are informative but limited, as they cannot determine whether an observed disease frequency is genuinely higher than expected. Incorporating a comparator cohort such as ADPKD allows calculation of incidence rate ratios (IRRs), strengthens causal inference by accounting for background event rates and temporal trends, and helps distinguish disease-specific risk from baseline risk in the broader ESKD population. This approach produces a more robust and clinically interpretable assessment of whether patients with pGNs experience excess risk compared with providing incidence estimates in isolation. In line 80 of the main manuscript, we cited a previous study that used the same approach to examine long-term outcomes in transplant patients, comparing those with pGNs to ADPKD.

2. It would be better to refer to APKD with the more formal name of “autosomal dominant polycystic kidney disease (ADPKD)”

Thank you for your comment. We have now adopted the full term “autosomal dominant polycystic kidney disease (ADPKD)” in the manuscript.

The language needs to be revised in several sections.

Introduction:

“Incidence of end stage kidney disease (ESKD) is steadily increasing worldwide, over a period of time (1, 2).” Over what period of time?

“Although considered rare in general population,” should be “in the general population”

“pGN as a group contribute” should be “contributes”

We appreciate these helpful corrections and will amend the manuscript accordingly, including:

• Clarifying the timeframe for increasing ESKD incidence. “Over the last two decades” (line 58 in main manuscript)

• “in the general population” (line 68 in main manuscript)

• “pGN as a group contributes…” (line 68 in main manuscript)

Methods:

The retrospective nature of the study is not pointed out in the methods.

Thank you for highlighting that. We have now clarified the retrospective nature of our study in the Methods section (line 96).

This was a retrospective cohort study using prospectively collected data from the UK Renal Registry. Patients were identified at the initiation of KRT, and outcomes were ascertained through routine registry linkage and annual reporting. We have clarified this in the Methods section.

“This was not recorded in substantial proportion and were treated as a separate category in the analyses.” Should be “in a substantial proportion” and “(this variable/ethnicity) was” “fitted survival rates and 1, 5, and 10 years” should be “fitted survival rates at 1, 5, and 10 years”

Thank you for your meticulous review and for highlighting these unintentional errors. Both have now been corrected in lines 150 and 185.

Results:

As the main aim is the incidence, it should be reported first.

We agree with the reviewer’s comments and have therefore added the incidence of kidney replacement therapy after the demographics section and before the patient survival outcomes.

The tables and supplemental material were not included in the provided manuscript file.

We note that the journal’s proofing system provided these as linked files. We have verified the upload and ensure all tables/supplementary files are clearly accessible in the resubmission package.

Discussion:

The link between migration and their impact on ethnicity in the UK therefore changing incidence of different types of pGN is not mentioned. Another factor that may have affected incidence and survival is the aging of populations.

We thank the reviewer and have added a brief comment in the Discussion (line 272-275) acknowledging demographic shifts, such as migration patterns and population ageing, as potential contributors to long-term changes in incidence and outcomes. We do, however, note that the incidence and survival models do adjust for age, sex and ethnicity.

Reviewer #2

General comments:

1. Thanks to the authors for the work done and the interest to share with the scientific world

We thank the reviewer for their thoughtful assessments.

2. The topic biases readers to think there is discussion of primary GNs and not sure why largely authors are rather comparing to ADPKD? Was rather thinking authors can look at comparing their major causes of GNs

We thank the reviewer for raising this point. The primary purpose of the study is indeed to describe the incidence and outcomes of the four major pGN subtypes individually and collectively, the manuscript reports these extensively in tables, figures, and supplementary material.

ADPKD is included in addition as a stable, well-recognised reference condition to contextualise pGN estimates, not as an alternative or competing primary focus. We have clarified this rationale in the Introduction and revised the terminology to avoid misinterpretation.

We have also referenced another UK study that used a similar approach to examine long-term post-transplant outcomes in patients with primary glomerular diseases (citation 8: Pruthi R, McClure M, Casula A, Roderick PJ, Fogarty D, Harber M, et al. Long-term graft outcomes and patient survival are lower posttransplant in patients with a primary renal diagnosis of glomerulonephritis. Kidney Int. 2016;89(4):918-26.).

3. Most of the comparisons are not done analytically. Looks mainly descriptive and it might help to know of these differences are statistically significant?

We respectfully disagree. The purpose of the study is descriptive epidemiology; the central question is “What are the incidence and survival estimates?” rather than “Are the groups statistically different?”.

Accordingly, we report:

a) effect sizes (incidence rate ratios, survival estimates), and

b) 95% confidence intervals

This, following modern best practice and avoiding reliance on P-values, consistent with – Gardner & Altman’s seminal argument for effect-size reporting (BMJ 1986), and recent recommendations in Nature and leading epidemiology journals.

All regression models are fully inferential. If a reader wishes to interpret significance, they may do so using the CIs provided. We have added brief clarification in the Methods to make this philosophy explicit.

Abstract:

1. Authors report regression models in the methods but just reports description in the results

We agree that the incidence reporting in the abstract was brief, and we have now expanded this section. The full abstract has been revised to address the reviewer’s comments.

Introduction:

1. Authors report IgA nephropathy (IgAN), membranous nephropathy (MN), focal segmental glomerulosclerosis (FSGS), and membranoproliferative glomerulonephritis (MPGN) as major causes of GN in the cohort and was rather expecting analysis of these and not ADPKD?

We don’t understand this remark and presume may be related to the reviewer not being able to view the tables. The results for the 4 conditions (and GN as a group) are extensively reported in the tables provided.

2. Might help not to start sentences with Arabic numerals.

We have revised all sentences beginning with Arabic numerals throughout the manuscript to conform with standard editorial style.

Methods:

1. Authors should report with some short notes on the UK renal registry for the benefit of the readership

We have added a dedicated paragraph in the Methods describing the UK Renal Registry’s prospective data collection, national population coverage, diagnostic coding framework, and data quality assurance processes ( from line 117-127).

2. Line 138 - Incidence rates are normalised to the relevant age, sex and country” – not sure if authors meant standardized?

This is now line 163. We have revised the wording to “standardised” to avoid ambiguity.

3. Line 159 - “Results are presented as fitted survival rates and 1, 5, and 10 years with 95% CI” – did not appreciate a lot of the 95% confidence interval the results.

The 95% confidence interval (CI) is the standard way to present the precision of estimates. We kindly request the reviewer to re-examine Tables 3–5 and Supplementary Tables 3, 4, and 6, where all results are reported with 95% CIs.

Results:

1. Line 169-170 “Most patients were white (89.1% in APKD and 78.7% in pGN). In the pGN group, a significant male predominance was observed at 71.6% compared to 53.1% in APKD” – was this proven to be statistically significant?

This is a comment on the demographic table 1 and describes the sample, statistical testing of differences is not appropriate here. If the reviewer is interested a trivial chi-squared test would be highly significant given the large numbers in the cohort. In the main analyses we do show the differences in incidence between the sexes and even comment that this is significant.

2. Authors should hope to report the significant of all the analysis statistically

We chose to use a standard epidemiological presentation which focuses on effect sizes and CI. This was for 2 reasons (a) to follow standard epidemiological practice and (b) to follow the movement across medical statistics and biosciences away from a P-value centric approach with the arbitrary P<0.05 threshold to a more nuanced effect-size approach. The lead statistician has hardly published a P-value in the last 10 years across a wide range of biomedical journals.

Discussion:

1. Might be helpful to have looked at the GNs as per topic and not sure why authors and would have loved to see how the various GNs progress and their transplant outcomes.

We thank the reviewer. Within the constraints of available UKRR data, we have fully reported transplant outcomes for all four pGN subtypes (Figures 4, 5, Tables 3, 5, supplementary figure 2). Natural-history analyses prior to ESKD (i.e., GN progression) are unfortunately not possible using this registry data, which is derived from ESKD registry, but we note this explicitly and mention ongoing follow-up for future analyses.

We believe all the GNs are fully reported in the tables, although some of the details of the 4 conditions are relegated to supplementary tables, given the extent of the data available and prioritised the more relevant ones in the main manuscript. Follow-up of this cohort is ongoing, and we would hope as the data matures to be able to add more on the disease progress, especially the transplant related outcomes.

---

## [Decision Letter · Decision Letter 1]

15 Jan 2026

Please ensure that each reviewer comment is thoroughly and accurately addressed.

Dear Dr. Ragy,

Thank you for submitting your manuscript to PLOS ONE. After careful consideration, we feel that it has merit but does not fully meet PLOS ONE’s publication criteria as it currently stands. Therefore, we invite you to submit a revised version of the manuscript that addresses the points raised during the review process.

We look forward to receiving your revised manuscript.

Kind regards,

Nasar Alwahaibi, PhD

Academic Editor

PLOS One

[This study was supported through a research grant from CSL Vifor.].

3. For studies involving third-party data, we encourage authors to share any data specific to their analyses that they can legally distribute. PLOS recognizes, however, that authors may be using third-party data they do not have the rights to share. When third-party data cannot be publicly shared, authors must provide all information necessary for interested researchers to apply to gain access to the data. (https://journals.plos.org/plosone/s/data-availability#loc-acceptable-data-access-restrictions)

5. Please include a copy of Table 1-5 which you refer to in your text on page 6-7.

7. Please include your tables as part of your main manuscript and remove the individual files. Please note that supplementary tables (should remain/ be uploaded) as separate "Supporting Information" files.

Additional Editor Comments (if provided):

Please ensure that each reviewer comment is thoroughly and accurately addressed.

Reviewers' comments:

Reviewer's Responses to Questions

**Comments to the Author**

Reviewer #3: (No Response)

Reviewer #4: (No Response)

2. Is the manuscript technically sound, and do the data support the conclusions?

Reviewer #3: Yes

Reviewer #4: Yes

3. Has the statistical analysis been performed appropriately and rigorously?

Reviewer #3: Yes

Reviewer #4: Yes

4. Have the authors made all data underlying the findings in their manuscript fully available?

Reviewer #3: Yes

Reviewer #4: Yes

5. Is the manuscript presented in an intelligible fashion and written in standard English?

Reviewer #3: Yes

Reviewer #4: Yes

Reviewer #3: The authors provide descriptive findings of a study using UK registry data to evaluate incidence rates of KRT and survival following KRT initiation by the different forms of primary glomerulonephritis and/or various demographic subgroups. I believe the authors have adequately addressed the comments of the reviewers from the first review, though I have a few additional comments:

Although PLOS One publishes epidemiologic studies of both a descriptive and etiologic nature, from research groups that publish one type of study more often than another (e.g. academic group, health agency), the wording in the early parts of the manuscript can explicitly state that it aims to describe incidence rates as a descriptive or surveillance study using national registry data, to make it more clear for readers who may be more used to etiologic studies with hypothesis testing.

The development of the two cohorts can also be made clearer, perhaps with a diagram akin to a STROBE diagram, but showing the timeline of the cohorts on an axis, with key points in time indicated such as the start and end of observation periods to ascertain initiation of KRT, mortality, left/right censoring, etc. I am not able to view supplementary material to see if this already exists, but I don't see a reference to a similar diagram in the text.

Reviewer #4: This study was based on national data from the UK Kidney Registry to analyze the epidemiological characteristics, incidence and survival outcomes of primary glomerulonephritis patients initiating renal replacement therapy from 1999 to 2019, with ADPKD as the control. The study design was reasonable, the sample size was large, the follow-up time was long, and the statistical methods were appropriate, which met the STROBE standards. The revised manuscript has complete structure, clear presentation, reliable conclusions, and clear clinical and public health value.

1. The methodological orientation of the control group was not clear

ADPKD is referred to as comparator in many cases, but it is not a matched control, nor is it an expose-non-exposure design. Its pathogenesis, disease progression, and transplantation strategy are significantly different from pGN. It is recommended that ADPKD be clearly defined as a reference disease in Methods, which is used to provide background incidence and outcome reference, rather than a causal comparison object. It is clearly pointed out in Discussion and Conclusions that the difference between pGN and ADPKD cannot be explained by the disease itself.

2. Insufficient confounding control

Studies had limited control for potential confounders in the main analysis. Comorbidities were used only in the subanalysis, ethnicity was severely missing and was not included in the analysis, and variables related to disease severity could not be adjusted, but the main outcome still presented a deterministic survival comparison. It is recommended that these limitations be acknowledged in the Discussion system, that differences in survival may be affected by selection by indication, and that the direction of main effect consistency may be reported in subsamples with complete comorbidity data to support the robustness of the results.

3. Follow-up and selection bias were not adequately emphasized

Only patients enrolled in KRT were included, which means that patients with mild or delayed pGN were missing, and the actual incidence was calculated as KRT-incidence rather than the overall population pGN incidence. It is recommended that this be clearly stated in the Abstract and Discussion to comply with STROBE requirements regarding study subject selection and interpretation of results.

4. PD and pre-emptive transplant conclusions need to be lowered

This article mentioned that PD has a similar performance to transplantation in IgAN, and pre-emptive transplantation has a better outcome, but the treatment selection bias is not fully controlled, and there is a risk of causal extrapolation. It is recommended to use conditional and exploratory language and to make it clear that the findings cannot be taken as evidence to guide treatment choices and need to be validated by prospective studies.

5. STROBE report completeness

The manuscript can further improve the STROBE report, and it is suggested to provide cohort flow diagram in supplementary materials and clarify the actual number of patients included in each analysis, especially in IMD and comorbidity subanalyses, to increase the transparency and reproducibility.

6. Handling of missing data

There are many missing key variables such as race. ethnicity can be used as the missing category, but the direction of potential bias should be explained, and a brief description of whether the correlation between the missing pattern and the outcome and its possible influence should be evaluated.

7. Statistical presentation

Some expressions can still be misleading; for example, "significant male predominance" is easily confused with statistical significance. In addition, for questions of multiple comparisons, such as incidence or survival between different pGN subtypes and ADPKD, adjustment should be stated in the methods or limitations, and if not, the justification should be explained.

8. Text and formatting problems

Some sentences in the manuscript still have redundancy, such as the first half of Introduction can be compressed; The values in the table are generally consistent with the text. Clear legends provided that all figures are correctly embedded with clear titles and notes; The reference format should be unified to ensure that the citations in the text are consistent with the references.

9. Consistency of key numbers

The total incidence of pGN in the abstract was "17.4 pmp", while the result of the text was "11.9 pmp" (line 247 of the text), which was inconsistent. The authors were asked to check the data and harmonize the figures to avoid confusing the readers.

10. Key findings and interpretation of clinical significance

Some of the results, such as the similar 5-year survival rate after PD and transplantation, or some differences in morbidity and survival, need to be interpreted with caution in the Discussion to avoid excessive inferences. Further exploration of potential biologic or clinical mechanisms for these unusual findings is warranted, and selection bias or unmeasured confounding may have influenced the results.

.

Reviewer #3: No

Reviewer #4: No

---

## [Author Response · Author response to Decision Letter 2]

24 Feb 2026

Summary of revisions

The following changes have been implemented or are to be actioned in the revised manuscript:

1. Abstract – KRT-incidence

a) We now used “incidence of KRT initiation due to primary glomerulonephritis rather than overall ESKD incidence.”

2. Introduction – descriptive intent

Added this sentence in Introduction: “This is a descriptive, population-based epidemiological study using national registry data, primarily designed to estimate the incidence of ESKD due to primary glomerulonephritis and associated long-term KRT outcomes, including modality use and patient and graft survival, rather than to test causal hypotheses.”

3. Methods – ADPKD as a reference disease (not a comparator)

a) Replaced ADPKD as the comparator with this wording: “ADPKD was included as a reference disease to provide contextual background for incidence and outcome estimates for a common, non–immune-mediated cause of ESKD with stable epidemiology; it was not intended as a matched control or exposure comparator, given its distinct pathogenesis, disease course, and transplantation pathways compared with primary glomerulonephritis.”

b) Added at the end of the Statistical Methods section: “Consistent with contemporary epidemiological reporting standards, analyses was focussed on estimation of effect sizes with corresponding confidence intervals rather than hypothesis testing based on P-values. Comparisons are descriptive and hypothesis-generating, and no formal adjustment for multiple comparisons was undertaken.”

4. Methods - missingness

a) Clarified: “Missingness was not strongly associated with calendar year or centre, although differential missingness by disease severity or socioeconomic factors cannot be excluded. Its potential impact, including for ethnicity, should be considered when interpreting subgroup estimates.”

5. Results / Abstract – incidence numbers

a) Added a sentence in Results to clarify: “The overall population incidence of KRT due to pGN was 11.8 per million population.”

6. Results – inferential language

a) Replaced “a significant male predominance was observed” with “A male predominance was observed.”

7. Discussion – PD and pre-emptive transplant conclusions

a) Replaced previous version with: “Peritoneal dialysis was associated with survival outcomes comparable to transplantation in selected subgroups, particularly IgAN, although these findings are observational and likely influenced by treatment selection and residual confounding. They should therefore be considered hypothesis-generating and require confirmation in prospective studies.”

8. Discussion – selection bias & confounding (in Limitations)

a) Expanded the limitations paragraph in Discussion and included the context: Several limitations inherent to registry-based observational studies should be considered when interpreting these findings. First, although the UK Renal Registry collects data prospectively, the present analyses were conducted retrospectively. Second, comorbidity data were available for approximately two-thirds of the cohort, and disease severity measures were not captured, limiting the ability to fully adjust for clinical risk. Third, primary renal diagnoses were based on registry coding (ERA–EDTA classifications) rather than central pathology review, and reasons for graft failure were unavailable. In addition, the MPGN category did not distinguish between primary complement-mediated disease and secondary causes such as infection or monoclonal gammopathy, which may introduce heterogeneity within this subgroup.

No formal adjustment for multiple comparisons was applied, as the analyses were descriptive and hypothesis-generating in nature. Residual confounding remains possible, including confounding by indication in dialysis modality selection and timing of transplantation. Although sensitivity analyses incorporating comorbidity data demonstrated consistent direction and magnitude of associations, incomplete covariate data and absence of detailed disease severity metrics limit causal interpretation. Accordingly, observed differences in survival should be interpreted as associations rather than evidence of treatment effect.

Furthermore, the registry captures only patients initiating kidney replacement therapy; individuals with pGN managed conservatively or with delayed progression are not included. Therefore, the reported incidence reflects KRT-incidence due to pGN rather than overall population incidence of disease. Despite these limitations, strengths of the study include nationwide coverage, high data completeness, minimal loss to follow-up, and the inclusion of ADPKD as a reference disease to contextualise incidence and outcome patterns.

9. Cohort flow / timeline diagram (Supplementary Figures)

a) Added cohort flow / timeline diagram in supplementary material (Supplementary Fig 4-7).

Editorial comments

Editorial comment 1

Authors’ response 1:

We confirm that the manuscript has been carefully reviewed and revised to comply fully with PLOS ONE’s style and formatting requirements.

Editorial comment 2

This study was supported through a research grant from CSL Vifor.

Authors’ response 2a:

Thank you for your comment. In Line 303 we stated the following under the funding section: “This study was supported through a research grant from CSL Vifor. The company had no role in the design of the study, analysis of the data, interpretation of the results, or in the preparation of the manuscript. The views expressed are solely those of the authors.”

Authors’ response 2b:

Thank you for your comment. We clarified with the statement in line 302 under the funding section.

Authors’ response 2c:

Thank you for your comment. None of the authors received a salary from the funder.

Authors’ response 2d:

Thank you for requesting clarification on the study funding. We have added a sentence in line 305.

Editorial comment 3

3. For studies involving third-party data, we encourage authors to share any data specific to their analyses that they can legally distribute.

PLOS recognizes, however, that authors may be using third-party data they do not have the rights to share. When third-party data cannot be publicly shared, authors must provide all information necessary for interested researchers to apply to gain access to the data. (https://journals.plos.org/plosone/s/data-availability#loc-acceptable-data-access-restrictions)

1) A description of the data set and the third-party source.

2) If applicable, verification of permission to use the data set.

3) Confirmation of whether the authors received any special privileges in accessing the data that other researchers would not have.

4) All necessary contact information others would need to apply to gain access to the data.

Authors’ response 3:

We thank the Editorial Board for the opportunity to clarify the data access arrangements.

- The data used in this study were obtained from the UK Renal Registry (UKRR) under a standard data access request, in accordance with UKRR governance procedures. The authors do not have legal authority to publicly distribute the individual-level registry data due to patient confidentiality and data protection regulations.

- The data underlying this study were obtained from the UK Renal Registry (UKRR), a national registry of patients receiving kidney replacement therapy in the United Kingdom. The dataset comprises de-identified individual-level data, including demographic characteristics, primary renal diagnosis (coded using ERA–EDTA classifications), treatment modality, transplantation events, and outcome data (including death and graft failure).

- We, as authors, accessed these data following approval of a standard data access application through the UK Renal Registry governance framework. Permission was granted for use of the dataset for the purposes of this specific research study.

- Due to legal, ethical, and data protection restrictions related to the UKRR policy, we are not permitted to publicly share the underlying individual-level data.

- We, as authors, confirm that we did not receive any special privileges in accessing the data that other qualified researchers would not have. Access is available to bona fide researchers through the UK Renal Registry’s formal data request process.

- Details regarding application procedures, governance requirements, and data access policies are available on the UKRR website. Researchers wishing to access the data may apply directly to the UK Renal Registry via: UK Renal Registry, UK Kidney Association

Editorial comment 4

Authors’ response 4:

Thank you for your comment. We have uploaded the figures again with a separate caption for each.

Editorial comment 5

5. Please include a copy of Table 1-5 which you refer to in your text on page 6-7.

Authors’ response 5:

Thank you for your comment. We have now included all five tables in the main manuscript as requested.

Editorial comment 6

Authors’ response 6:

Thank you for highlighting this point. We have now added captions for all supplementary tables/figures and uploaded as a separate file.

Editorial comment 7

7. Please include your tables as part of your main manuscript and remove the individual files. Please note that supplementary tables (should remain/ be uploaded) as separate "Supporting Information" files.

Authors’ response 7:

Thank you for your comment. We have now included all five tables in the main manuscript as requested and removed the individual files.

Responses to Reviewers’ Comments

Reviewer #3:

The authors provide descriptive findings of a study using UK registry data to evaluate incidence rates of KRT and survival following KRT initiation by the different forms of primary glomerulonephritis and/or various demographic subgroups. I believe the authors have adequately addressed the comments of the reviewers from the first review, though I have a few additional comments.

Authors’ response: We thank the reviewer for their careful re-evaluation of the manuscript and for recognising the revisions made in response to the initial review. We appreciate their positive assessment and are pleased that the clarifications and additional revisions have addressed the previous concerns.

1. Although PLOS One publishes epidemiologic studies of both a descriptive and etiologic nature, from research groups that publish one type of study more often than another (e.g. academic group, health agency), the wording in the early parts of the manuscript can explicitly state that it aims to describe incidence rates as a descriptive or surveillance study using national registry data, to make it more clear for readers who may be more used to etiologic studies with hypothesis testing.

Authors’ response: We thank the reviewer for this constructive suggestion. We agree that clearer framing will benefit readers who may be more familiar with etiologic or hypothesis-driven studies. We have revised the Introduction to explicitly state (lines 79-82) “This is a descriptive, population-based epidemiological study using national registry data, primarily designed to estimate the incidence of KRT due to primary glomerulonephritis and associated long-term KRT outcomes, including modality use and patient and graft survival, rather than to test causal hypotheses.”

2. The development of the two cohorts can also be made clearer, perhaps with a diagram akin to a STROBE diagram, but showing the timeline of the cohorts on an axis, with key points in time indicated such as the start and end of observation periods to ascertain initiation of KRT, mortality, left/right censoring, etc. I am not able to view supplementary material to see if this already exists, but I don't see a reference to a similar diagram in the text.

Authors’ response: We thank the reviewer for this helpful suggestion. To improve clarity and transparency, we have added a schematic diagram illustrating the construction of the incidence and survival cohorts, including the observation periods, censoring, and key time points for KRT initiation, mortality, and follow-up (Supplementary Figure 4, 5 and 6). These diagrams are provided as supplementary figures in supplementary material and referenced in the patient’s cohort and definition sections.

Reviewer #4:

This study was based on national data from the UK Kidney Registry to analyze the epidemiological characteristics, incidence and survival outcomes of primary glomerulonephritis patients initiating renal replacement therapy from 1999 to 2019, with ADPKD as the control. The study design was reasonable, the sample size was large, the follow-up time was long, and the statistical methods were appropriate, which met the STROBE standards. The revised manuscript has complete structure, clear presentation, reliable conclusions, and clear clinical and public health value.

Authors’ response: We sincerely thank the reviewer for their thoughtful evaluation of our work and for recognising the strengths of the study design, dataset, and analytical approach. We appreciate their positive assessment of the manuscript’s clarity, methodological rigor, and clinical and public health relevance. We are grateful for the time and expertise invested in reviewing our work.

1. The methodological orientation of the control group was not clear ADPKD is referred to as comparator in many cases, but it is not a matched control, nor is it an expose-non-exposure design. Its pathogenesis, disease progression, and transplantation strategy are significantly different from pGN. It is recommended that ADPKD be clearly defined as a reference disease in Methods, which is used to provide background incidence and outcome reference, rather than a causal comparison object. It is clearly pointed out in Discussion and Conclusions that the difference between pGN and ADPKD cannot be explained by the disease itself.

Authors’ response: We thank the reviewer for this important clarification. We agree and have revised the manuscript methodology section (lines 93-97) to consistently define ADPKD as a reference disease, rather than a matched control or exposure group. Its role is to provide background incidence and outcome context for a common, non-immune mediated cause of ESKD with stable epidemiology, rather than to imply causal comparability with pGN.

2. There was limited control for potential confounders in the main analysis. Comorbidities were used only in the sub analysis, ethnicity was severely missing and was not included in the analysis, and variables related to disease severity could not be adjusted, but the main outcome still presented a deterministic survival comparison. It is recommended that these limitations be acknowledged in the Discussion system, that differences in survival may be affected by selection by indication, and that the direction of main effect cons

---

## [Decision Letter · Decision Letter 2]

26 Mar 2026

ESKD from Primary Glomerulonephritis: Incidence and post KRT survival: A national cohort study in the UK

PONE-D-25-50346R2

Dear Dr. Ragy,

We’re pleased to inform you that your manuscript has been judged scientifically suitable for publication and will be formally accepted for publication once it meets all outstanding technical requirements.

Kind regards,

Nasar Alwahaibi, PhD

Academic Editor

PLOS One

Additional Editor Comments (optional):

Reviewers' comments:

Reviewer's Responses to Questions

**Comments to the Author**

Reviewer #1: All comments have been addressed

2. Is the manuscript technically sound, and do the data support the conclusions?

Reviewer #1: Yes

3. Has the statistical analysis been performed appropriately and rigorously?

Reviewer #1: Yes

4. Have the authors made all data underlying the findings in their manuscript fully available?

Reviewer #1: Yes

5. Is the manuscript presented in an intelligible fashion and written in standard English?

Reviewer #1: Yes

Reviewer #1: (No Response)

.

Reviewer #1: **Yes:** Yasmine NagaYasmine NagaYasmine NagaYasmine Naga

---

## [Editor Report · Acceptance letter]

PONE-D-25-50346R2

PLOS One

Dear Dr. Ragy,

I'm pleased to inform you that your manuscript has been deemed suitable for publication in PLOS One. Congratulations! Your manuscript is now being handed over to our production team.

Kind regards,

on behalf of

Dr. Nasar Alwahaibi

Academic Editor

PLOS One